# Chinese Named Entity Recognition Based on BERT and Lightweight Feature Extraction Model

**Ruisen Yang** [1] , **Yong Gan** [2,*] **and Chenfang Zhang** [1]

1 School of Computer Communication and Engineering, Zhengzhou University of Light Industry, Zhengzhou 450000, China
2 School of Computer Communication and Engineering, Zhengzhou Institute of Engineering and Technology, Zhengzhou 450000, China
* Correspondence: ganyong@zzuli.edu.cn

**Abstract:** In the early named entity recognition models, most text processing focused only on the representation of individual words and character vectors, and paid little attention to the semantic relationships between the preceding and following text in an utterance, which led to the inability to handle the problem of multiple meanings of a word during recognition. To address this problem, most models introduce the attention mechanism of Transformer model to solve the problem of multiple meanings of a word in text. However, the traditional Transformer model leads to a high computational overhead due to its fully connected structure. Therefore, this paper proposes a new model, the BERT-Star-Transformer-CNN-BiLSTM-CRF model, to solve the problem of the computational efficiency of the traditional Transformer. First, the input text is dynamically generated into a character vector using the BERT model pre-trained in large-scale preconditioning to solve the problem of multiple meanings of words, and then the lightweight Star-Transformer model is used as the feature extraction module to perform local feature extraction on the word vector sequence, while the CNN-BiLSTM joint model is used to perform global feature extraction on the context in the text. The obtained feature sequences are fused. Finally, the fused feature vector sequences are input to CRF for prediction of the final results. After the experiments, it is shown that the model has a significant improvement in precision, recall and F1 value compared with the traditional model, and the computational efficiency is improved by nearly 40%.

**Keywords:** named entity recognition; deep learning; neural network; BERT





## 1. Introduction

In the age of information technology, people receive all kinds of data every day, which have the characteristics of no structure, among which, text type data occupies a significant proportion. In the face of huge amounts of data, it is not possible to process these contents by manual work alone, so the concept of named entity recognition (NER) has been proposed to process text by computer.

NER is a subtask of information extraction that aims to locate and classify named entities in text into pre-defined categories such as people, organizations, locations, time expressions, quantities, monetary values, percentages, etc. Named entity recognition is one of the hot research directions in natural language processing, aiming at identifying named entities in text and grouping them into the corresponding entity types. Named entity recognition is a very fundamental task in natural language processing (NLP), and is an important basic tool for many NLP tasks such as information extraction, question and answer systems, syntactic analysis, and machine translation [1].

In the early named entity recognition, the lexicon and rule-based approach is the mainstream approach in NER. The core idea of this approach is to use external lexicon and input data for matching to achieve the effect of classification, but this approach can only

have a high effect in a specific text, and when new named entities appear, the lexicon has to be updated, which is not only laborious but also low in portability, and less applicable in dealing with complex texts in other fields.

With the rise of machine learning in the field of NLP, the use of machine learning methods for NER tasks has become a new trend [2]. In this trend, how to better solve the sequence annotation problem becomes the key to improving named entity recognition. However, this method has high requirements regarding feature selection, which not only requires selecting various features from the text that have an impact on the task to be added to the feature vector, but also requires selecting the set of features that can effectively reflect the characteristics of a particular named entity based on the characteristics exhibited by that entity recognition, which leads to its poor generalization ability.

In recent years, deep learning approaches have become mainstream, and various word vector representations are becoming more and more diverse, and neural network-based approaches have made greater progress in NER, such as with typical serialized annotation tasks.

## 2. Related Work

In early named entity recognition tasks, the main goal was to automatically identify named entities from a pile of textual data. A system for automatic recognition of company names was introduced by Rau [3] and others at the 1991 Conference on Applications of Artificial Intelligence, where the main approach used heuristics and manual rules. In 1996, the term named entity recognition was formally introduced by R. Grishman and Sundheim [4] at the MUC-6 conference, which led to an increasing interest in the field, and thus it entered a period of rapid development.

In recent years, deep learning techniques based on neural network models have become mainstream in NER tasks due to the rapid development of deep learning techniques, and the features of the method that do not rely on manual feature extraction have become an important reason for this being the main choice. The neural network-based method simply converts text information into vector form by learning the embedded model, and then the text information represented by the vector form is input into the neural network model, and the neural network encodes the information by modeling the text sequence, and finally decodes it in the decoding layer to obtain the final annotated sequence [5].

Currently, among the generative models of word vectors, there is the Word2Vec model proposed by Mikolov et al. [6]. This is a method of word embedding, which allows the natural language space and mathematical space to be connected. Based on the previous work, Pennington et al. proposed the Glove model [7]. One of the main advantages of Glove is that it uses uniformly distributed squared loss, which is better adapted to the cross-entropy loss function and has better expressiveness, and from the point of view of model training, Glove can obtain a more reliable set of word vectors faster. However, both models produce the same word vectors in the corpus of the text, which leads to the problem of multiple meanings in both of them, which leads to an impact on the results of subsequent tasks. Devlin et al. [8], in 2018, proposed the BERT (bidirectional encoder representation from transformers) model, which has been proven after extensive experiments to have better performance in pre-trained language models for NLP, which can capture the features of long texts and can dynamically generate word vectors in different contexts with better computing power, and has become the main pre-processing model in the field of NLP.

In sequence annotation tasks, the commonly used encoding methods are RNN (recurrent neural networks), and variants of RNN, LSTM [9] (long short-term memory) and CNN [10,11] (convolutional neural networks). The first feature extractor that was used for named entity recognition tasks was CNN, and the algorithm achieved good recognition results in the image domain, so researchers started using CNN for NLP. CNNs are effective in extracting local features by using a convolution kernel of the same dimension as the character vector to convolve with a matrix composed of character vectors. The advantage of CNN networks is that in addition to local extraction, the parallelism of GPUs can be

used to extract local features quickly, but the disadvantage is that it is difficult to ensure that the extracted character features contain global information. However, even so, CNN is still a very effective processing model. Collobert and Weston et al. [12]. First proposed the application of convolutional neural networks in natural language processing in their 2008 publication, proposing that each input word corresponds to a word vector. Collobert et al. [13] proposed a generalized CNN framework in 2011, and using this generalized CNN framework, many NLP-type problems can be solved. Inspired by Collobert, many scholars started to study CNNs more deeply. However, due to the shortcomings of CNN in feature extraction of global information, academics proposed the RNN model. RNN cuts the input information into multiple tasks, and the output at a certain time is not only related to this input but also to the output of the previous time series. For each input of temporal information, it is sent to a place called a recurrent neural unit and then outputs a vector with pre-set dimensions [14]. Because the output of a recurrent neural unit is not only related to the input but also to the output of the previous recurrent unit, people figuratively call recurrent neural networks memorable. Once the recurrent neural network was introduced, it achieved good results. As researchers studied RNNs more deeply, it was found that RNNs gradually lose their learning ability as the sequence length increases, and the problem of 'gradient disappearance' and gradient explosion will occur. To address this problem, scholars have improved the traditional RNN and obtained an improved version of RNN, LSTM [15]. LSTM adds a 'gate' structure to the recurrent neural network, which controls the input at each time node and solves the gradient disappearance and gradient explosion problems to some extent. The problem of in-paragraph utterance is that the before-and-after relationship of text is often correlated, but LSTM can only capture text features in a single direction, thus ignoring the before-and-after relationship of the utterance, therefore, the BiLSTM model has been proposed [16]. BiLSTM does not change much from LSTM in structure, and BiLSTM can capture features from both directions of text, which greatly improves the recognition rate of named entities. Although RNNs have more efficient feature extraction capability compared to CNNs, the inability of RNNs to use GPUs for parallelism computation leads to RNNs being less efficient than CNNs in terms of usage. In 2017, Google proposed the Transformer encoder model with more powerful corpora extraction capability [17]. Each word in the Transformer encoder is compared to the other words in the sentence, the attention of the other words in the sentence is calculated, so that the true attention weight of the words in the whole sentence can be calculated more accurately. It has been experimentally demonstrated that the Transformer model achieves better results in NLP tasks, but because the structure of the Transformer model is fully connected, its computational and memory overheads are squared by a number of times determined by the sentence length, and the reference volume is larger, requiring a longer training time.

In the decoding stage, the commonly used models are SoftMax [18] and the CRF model [19], among which thenCRF model is the most classical model to solve the sequence labeling problem. In the entity recognition task, the input is a sentence text, and if the correlation information of the upper neighboring tags can be used to decode the best prediction result, the CRF model takes the relationship between the tags and the preceding and following text annotations into full consideration, so it can better solve the sequence annotation problem.

Since LSTM deals with the problem of gradient disappearance and gradient explosion occurring in time series data and can capture and preserve the contextual relationship of sequences well, the LSTM-CRF model has now become one of the basic network frameworks for NER tasks, and many scholars improve on it to improve the recognition of named entities. For example, Lample et al. [20], in 2016, proposed the BiLSTM-CRF model, which performs feature extraction in the front and back directions of the text to ensure the connection between contextual features, and thus obtained a more desirable recognition effect at that time. Huang et al. [21] added manual spelling features to BiLSTM-CRF in order to enrich its input feature representation. Wu et al. [22] proposed a CNN-LSTM-CRF

model to obtain short- and long-range content dependencies, and proposed to jointly learn NER and word separation tasks and explore the intrinsic connection between these two tasks. Santo et al. [23] added a convolutional layer to the CNN-CRF model to extract character-level features. Strubell et al. [24] first proposed a null convolutional network (IDCN) to extract features, which expands the perceptual field of view with a reduced number of parameters. However, the model with CNN as the basic structure cannot fully obtain the global information, so the recognition effect is lower than the joint model of CNN and RNN.

Since BERT can fully characterize syntactic and semantic information in different contexts, the BERT model has started to be used as a preprocessing model for solving the problem of multiple meanings of words. For example, Straková et al. [25] applied the BERT model to nested named entity recognition and improved the recognition effect. Gao et al. [26] used the BERT-BiLSTM-CRF model for named entity recognition in Chinese and in the publicly released dataset of EMR named entity recognition evaluation task in the CCKS2020 competition, the named entity recognition for drug type entities reached an F1 value of 96.82%.

In recent years, after the continuous efforts of researchers, the recognition of named entities has become more and more effective, so the method of named entity recognition has been applied to some specific fields. For example, Liu et al. [27] used named entity recognition for named entities in geological texts. Yang et al. [28] used a medical named entity recognition method with weakly supervised learning to perform experiments on the CCKS2017 official test set. The experimental results show that the weakly supervised learning methods proposed in this paper achieve the satisfactory performance as the supervised methods under comparable conditions. Zhuang et al. [29] used a model based on the BiLSTM method to apply the named entity recognition task to the field of journalism.

## 3. Model Introduction

### 3.1. BERT-Star-Transformer-CNN-BiLSTM-CRF Model

The model used in this paper is mainly divided into three parts: word vector embedding layer, feature extraction and feature fusion layer, and decoding layer, and the overall framework of the model is shown in Figure 1. The word vector embedding layer is mainly a BERT model, which is also the core part of this model, aiming to solve the problem of multiple meanings of words in text. The feature extraction and feature fusion layer mainly consist of three models, Star-Transformer, CNN and BiLSTM, with Star-Transformer and CNN models to capture local features and the BiLSTM model to capture global features. The CRF model is used for the decoding layer.

The model first encodes the input text using the BERT pre-trained language model to obtain a sequence of word vectors, and then inputs them into the joint Star-Transformer-CNN-BiLSTM model to extract feature information, and the joint model fuses the obtained features to obtain a new sequence of vectors, and finally inputs the fused feature vectors into the CRF layer to obtain each character's tag category. After experiments, it is proven that the model used in this paper has good performance in recognizing the efficiency of named entities based on solving the multiple meanings of a word. The structure of the proposed model in this paper is shown in Figure 1.

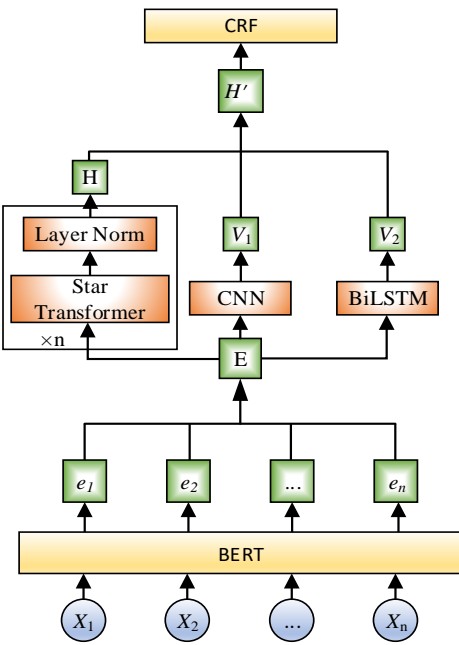

**Figure 1.** Overall framework of the model.

### 3.2. BERT Model

Word embedding techniques are designed to map words in natural language into a continuous vector space that can be understood by a computer, thus enabling the computer to process textual information. However, traditional word embedding learns the fixed semantics of a word and cannot solve the problem of multiple meanings of a word. To address this problem, this paper adopts the Chinese BERT pre-training model released by Google. The structure of the BERT model is shown in Figure 2.

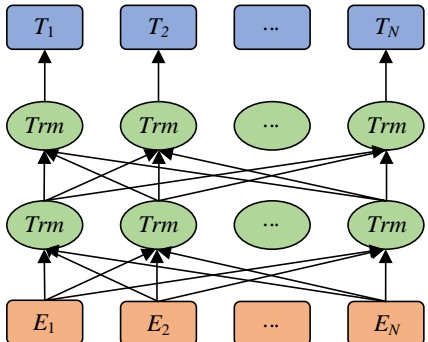

**Figure 2.** BERT model structure.

Bert is based on the transformer encoder model with a deep bidirectional network structure. Transformer encoder consists of multiple units stacked consecutively, where each unit has the same composition structure and consists of a multi-headed attention mechanism, normalization process, residual linking and feedforward neural network. By this method, the BERT model obtains different vector representations of the same vector for different semantic environments of sentences, thus solving the problem of multiple meanings of a word. The structure of the transformer encoder unit is shown in Figure 3.

The input of Bert consists of three parts, segment embeddings, token embeddings and positional embeddings, where segment embeddings are segmentation vectors, token embeddings are character vectors, and positional embeddings are used for downstream classification tasks. Positional embeddings are position vectors, which are used to obtain the relative position information of each character in the sequence.

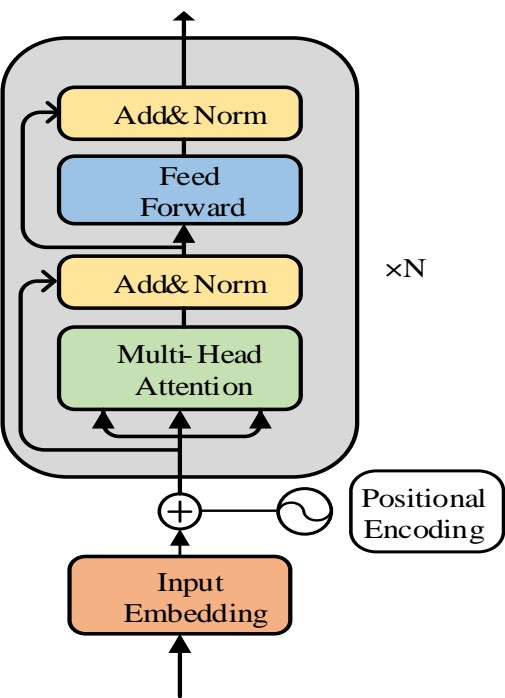

**Figure 3.** Transformer encoder unit.

There are several versions of the trained BERT model (https://github.com/google-research/bert, accessed on 2 September 2022), as shown in Table 1.

**Table 1.** BERT version information.

| Version | Applicable Language | Number of Layers | Number of Neurons | Number of Attention Mechanism Heads | Number of Parameters |
|---|---|---|---|---|---|
| BERT-Base | Chinese | 12 | 768 | 12 | 110 M |
| BERT-Base | English is not case sensitive | 12 | 768 | 12 | 110 M |
| BERT-Large | English is not case sensitive | 24 | 1024 | 16 | 340 M |
| BERT-Base | English is case sensitive | 12 | 768 | 12 | 110 M |
| BERT-Large | English is case sensitive | 24 | 1024 | 16 | 340 M |
| BERT-Base | Multilingual and case sensitive | 12 | 768 | 12 | 110 M |
| BERT-Base | Multilingual case sensitive | 12 | 768 | 12 | 110 M |

### 3.3. Star-Transformer Model

The Transformer model is popular in NLP applications because of its excellent attention mechanism. Transformer can replace recurrent neural networks and convolutional neural networks in many NLP tasks, such as GPT, BERT and Transformer-XL. However, the structure of Transformer leads to it having two limitations:

(1)  The fully connected structure of Transformer leads to a model with computational and memory overheads that are squared by a number of times according to the length of the sentence, which results in a longer training time for the model when the sentence is too long.

(2)  Transformer can only handle fixed-length text strings. The text must be split into a certain number of paragraphs or chunks before it can be entered into the system, and such chunks of text can lead to context fragmentation. For example, if a sentence is separated from the middle, then a large amount of context is lost.

In order to solve the above problems, this paper adopts a lightweight Star-Transformer model to extract sentence features. The core idea of the Star-Transformer model is to convert the fully connected structure into a star structure, and solve the problem of computational efficiency and context information loss by reducing the original connection structure [30]. The structure diagram of the Star-Transformer model is shown in Figure 4.

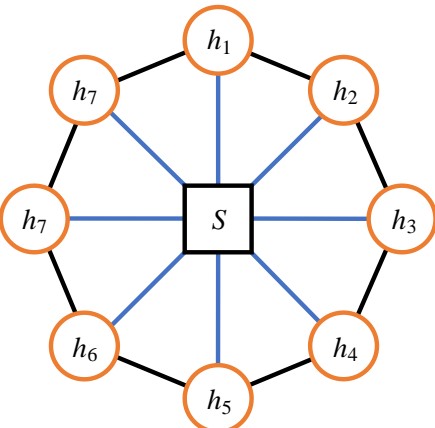

**Figure 4.** Star-Transformer model structure.

The Star-Transformer model is a star shape from the structural point of view. The nodes around are satellite nodes, and each satellite node represents the characteristics of a character. In the center of the model is the relay node, which receives information from all satellite nodes and sends messages to all satellite nodes.

Two types of links exist in the Star-Transformer model:

1. Ring connection: the satellite nodes are connected in a ring, and each satellite node is connected to the central node.
2. Basic connection: each satellite node is connected to a relay node.

The ring connection serves to obtain local information around each satellite node, while the basic connection allows the satellite nodes to obtain non-local information in the relay nodes.

The update process of satellite nodes and relay nodes is as follows:

Suppose the length of the text sequence is $n$, then the character embedding sequence is: $E = [e_1, e_2..., e_n]$, where each character vector is the state of the satellite node.

Setting the character dimension to $d$, the state of the relay node in the Star-Transformer model is $s^t \in R^{1 \times d}$ and the state of the $n$ satellite nodes is $H^t \in R^{n \times d}$ after the $t$-step update. Initialize $H^0 = E$, $s^0 = $ average $(E)$.

### 3.3.1. Attention Mechanism

The attention mechanism was first used in the image domain, and in 2014, Mnih et al. [31] added the attention mechanism to image classification tasks to improve the classification results. Xu et al. [32] introduced the attention mechanism to describe the image content. Thus, some scholars started to apply the attention mechanism to NLP tasks.

The attention mechanism involves three vectors: the query vector $Q$, the key-value vector $K$ and the value vector $V$. The role of vector $V$ is to store the information to be extracted, while vector $Q$ and vector $K$ work together to determine the importance of the information and thus the proportion of information to be extracted. Finally, the obtained proportion is weighted and summed with the vector $V$ to obtain the valuable information. The structure of the attention mechanism model is shown in Figure 5.

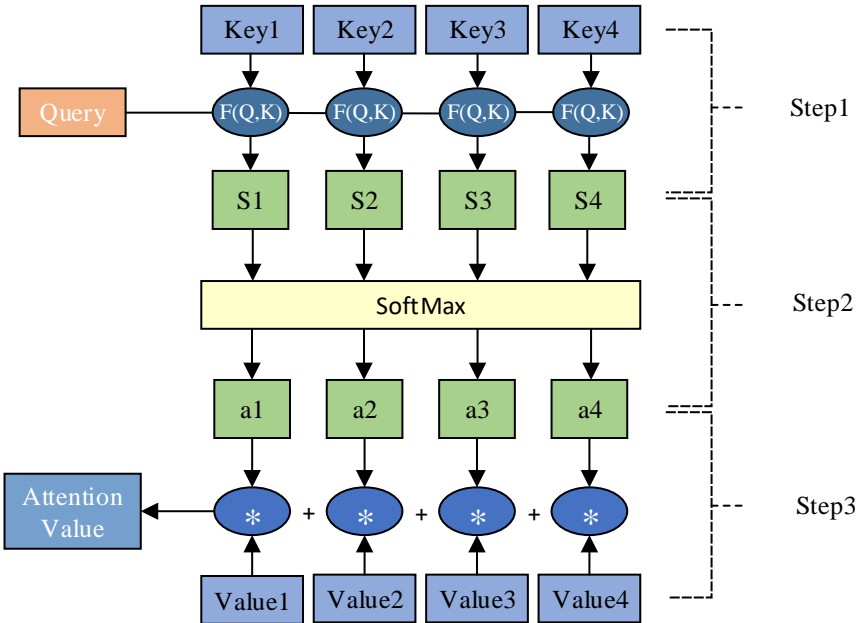

**Figure 5.** Attention mechanism model structure.

The calculation process of the attention mechanism is shown as follows:

1. Calculate the weight coefficients based on vector *Q* and vector *K* to obtain the similarity scores of both. There are methods to calculate the similarity such as vector dot product (Equation (1)) and cosine similarity (Equation (2)).

$$s_i(q, k_i) = q^T k_i \tag{1}$$

$$s_i(q, k) = \frac{q^T k_i}{\|q\| \cdot \|k_i\|} \tag{2}$$

2. Use the SoftMax function to perform the normalization operation on the similarity scores calculated in the previous step. The specific procedure is shown in Equation (3).

$$\alpha_i = \frac{\exp(S_i)}{\sum\limits_{j=1}^{N} \exp(S_j)} \tag{3}$$

3. Finally, the weight coefficients are multiplied with the vector *V* for weighted summation, and finally the Attention value against *Q* is obtained. The specific operation process is shown in Equation (4).

$$\text{Attention}((K, V), Q) = \sum_{i=1}^{N} \alpha_i v_i \tag{4}$$

### 3.3.2. Multiple Attention Mechanism

In contrast to the attention mechanism, the multi-headed attention mechanism can focus on information from different levels and on different subspaces. The use of the multi-headed attention mechanism was first proposed in Transformer, which uses multiple sets of parameter matrices *Q*, *K*, and *V* to linearly transform the vectors by stitching the information obtained from each head and converting the stitched matrix into a new vector, as shown in Equations (5) and (6):

$$Multihead = concat\ (\ head_1, \cdots,\ head_n) \cdot W \tag{5}$$

$$head_i = Attention\left(QW_q^i, KW_k^i, VW_v^i\right) \tag{6}$$

### 3.3.3. Update of Satellite Node

The state of each satellite node $h_i$ is updated by performing multi-head attention with its neighboring nodes, including the neighboring nodes $h_{i-1}$, $h_{i+1}$ in the sequence, the relay node $s$, and the previous state $h_i^{t-1}$ of the $h_i$ node with its corresponding character embedding $e_i$, and the update process is shown in Equations (7) and (8):

$$C_i^t = \left[h_{i-1}^{t-1}; h_i^{t-1}; h_{i+1}^{t-1}; e_i; s^{t-1}\right] \tag{7}$$

$$h_i^t = MultiAtt\left(h_i^{t-1}, C_i^t\right) \tag{8}$$

where $C_i^t$ is the contextual information of the i-th satellite node, so that the update of each satellite node is similar to a recursive network and the update is based on the attention mechanism. After the information exchange, the state of that node is calculated using layer normalization, and the calculation process is shown in Equation (9):

$$h_i^t = \text{LayerNorm}\left(\text{ReLU}\left(h_i^t\right)\right), i \in [1, n] \tag{9}$$

### 3.3.4. Update of Relay Node

The relay node obtains the global information by performing multi-headed attention with all satellite nodes and their previous states, and the update process is shown in Equations (10) and (11):

$$s^t = MultiAtt\left(s^{t-1}, \left[s^{t-1}; H^t\right]\right) \tag{10}$$

$$s^t = \text{LayerNorm}\left(\text{ReLU}\left(s^t\right)\right) \tag{11}$$

After each update is performed, the satellite nodes and relay nodes achieve a new state, and the local and non-local information in the sentence can be obtained from the node states after the $t$-step update.

The overall update process of Star-Transformer is shown in Algorithm 1.

---

**Algorithm 1.** The Update of Star-Transformer.

---

Input: $E = [e_1, e_2, \cdots, e_n]$
Output: $H = [h_1, h_2, \cdots, h_n]$
1 //*Initialization*
2 $\mathbf{h}_1^0, \cdots, \mathbf{h}_n^0 \leftarrow \mathbf{e}_1, \cdots, \mathbf{e}_n$
3 $s^0 \leftarrow$ average$(e_1, \cdots, e_n)$
4 for t 1 to T do
5 　//*update the satellite*
6 　for i 1 to n do
7 　　$\mathbf{C}_i^t = \left[\mathbf{h}_{i-1}^{t-1}; \mathbf{h}_i^{t-1}; \mathbf{h}_{i+1}^{t-1}; \mathbf{e}_i; \mathbf{s}^{t-1}\right]$
8 　　$\mathbf{h}_i^t = $ MultiAtt $\left(\mathbf{h}_i^{t-1}, \mathbf{C}_i^t\right)$
9 　　$\mathbf{h}_i^t = $ LayerNorm $\left(\text{ReLU}\left(\mathbf{h}_i^t\right)\right), i \in [1, n]$
10 　//*update the relay node*
11 　$\mathbf{s}^t = $ MultiAtt$\left(s^{t-1}, \left[s^{t-1}; \mathbf{H}^t\right]\right)$
12 　$s^t = $ LayerNorm$\left(\text{ReLU}\left(s^t\right)\right)$

---

### 3.4. CNN Model

In the image field, CNN has been widely used in the local information extraction of images. CNN in images is combined by one or more build layers, and the convolutional layer consists of multiple sets of convolutional kernels, and one convolutional layer contains a set of convolutional kernels. In an image, the convolutional kernels learn the local features of the image by sliding the relationships between the data in a top-to-bottom, left-to-right order on the two-dimensional data of the image to extract the local features. The structure of the CNN model in the image domain is shown in Figure 6.

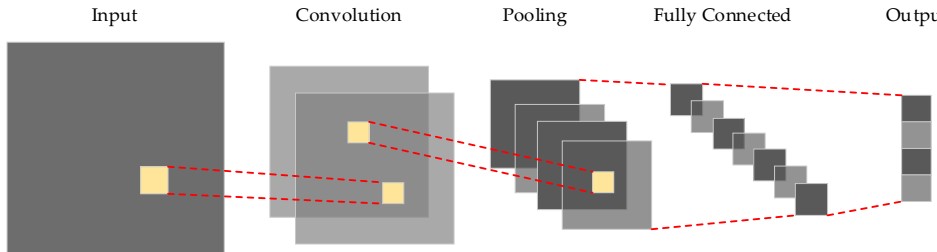

**Figure 6.** CNN model structure.

Mostly, research has shown that CNNs are very effective in extracting local features [33], so CNNs started to be used in NLP tasks. However, the CNN used in text is different from the CNN in image, image is two-dimensional data, while text is one-dimensional data, although the text has been mapped to two-dimensional data after the word embedding layer, it is meaningless to convolve the text vector from left to right, so the convolution of text only needs to be performed in one direction from top to bottom. The specific process is that the input $C_1$, $C_2$, $C_3$,....$C_N$ is mapped to vector form, and assuming the vector dimension size is $d$, an $n \times d$ matrix can be obtained, and the CNN is used to do a one-dimensional convolution on it. Let $w$ be a convolution kernel of CNN with dimension $a \times d$ and $a$ be the window size. By convolving the kernel with the matrix inside the window, the contextual representation $x_j^{CNN}$ of the $j$-th word is obtained, as expressed in Equation (12):

$$x_j^{CNN} = f\left(w^T x_{[j-\frac{a-1}{2}];[j+\frac{a-1}{2}]}\right) \tag{12}$$

where $x_{[j-\frac{a-1}{2}];[j+\frac{a-1}{2}]}$ denotes a window matrix of size a $\times$ a stitched by the word vectors in the $\left[j - \frac{a-1}{2}\right]$-th and $\left[j + \frac{a-1}{2}\right]$-th rows of the matrix, and $f$ is the nonlinear activation function, and the ReLU activation function is used here, as shown in Equation (13):

$$f(x) = \max(0, x) \tag{13}$$

In addition, in order to make the input dimension of CNN consistent, padding operation is usually applied to the input subsequence. In the actual network design, padding is usually set to SAME.

Each convolution kernel performs one convolution operation to obtain a new feature representation, Assuming $N = 5$ and $d = 5$, the input is a $5 \times 5$-word vector with a window matrix $x_{j-1:j+1}(j \in [1,5])$, and let $a = 3$, the size of the convolution kernel $W$ is $3 \times 5$. After convolving $x_{0:2}, x_{1:3}, x_{2:4}, x_{3:5}, x_{4:6}$ with w in turn by Equation (13), the new feature representation corresponding to the input word sequence can be obtained. The text sequence convolution process is shown in Figure 7.

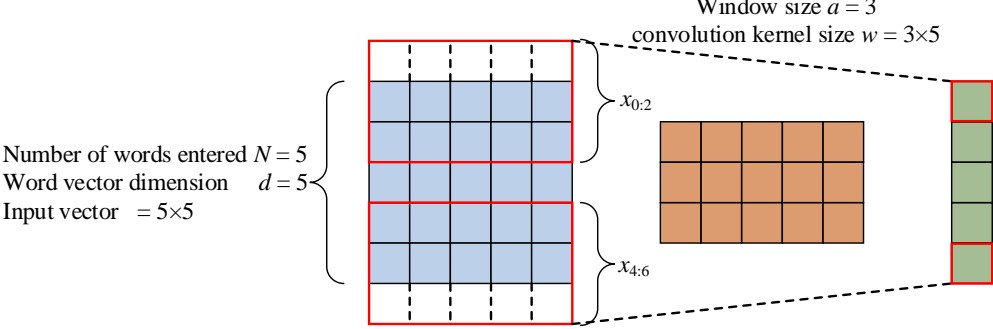

**Figure 7.** Text sequence convolution process.

By defining convolutional kernels with different window sizes, the relationships between different ranges of text can be learned. In defining multiple convolutional kernels

for each size window, complementary features are learned in the same window, so that more different local contextual features can be extracted.

### *3.5. BiLSTM Model*

### 3.5.1. LSTM Model

CNN cannot capture global information in feature extraction task, while RNN has a good extraction effect on the global information of feature sequences, so RNN started to be used in NLP tasks. Traditional RNNs have good sequence modeling capability, but when acquiring feature information over long distances, gradient vanishing and gradient explosion problems occur due to their own structure. For example, when the activation function in the network is sigmoid, the maximum value of its derivative is 0.25. With the stacking of neural network layers, the cumulative multiplication of multiple numbers less than or equal to 0.25 will lead to an operation result close to 0. Therefore, the parameters basically do not change when the gradient is propagated to the shallow network, which leads to the inability of the network to be trained, i.e., the gradient vanishes. To address these problems, researchers have improved the RNN by proposing a long short-term memory (LSTM). LSTM uses a gate control approach to structurally improve the RNN. The structure of the LSTM is shown in Figure 8.

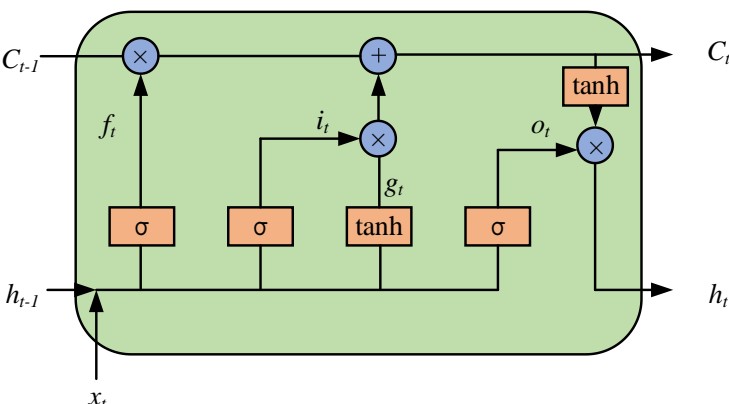

**Figure 8.** LSTM model structure.

LSTM adds three gate structures, input gates, forgetting gates and output gates, to the traditional RNN. Among them, the input gate is used to control the degree of information input into the state unit, the forgetting gate is used to filter the previously input information and control the proportion of retained information, and the output gate is used to determine the value of the next hidden state, which contains the previously input information.

When the LSTM is processing a sequence of features of a text, a moment $t$ corresponds to a word vector $x_t$ in the sequence. The input information of the LSTM contains the current word vector $x_t$, the hidden output $h_{t-1}$ of the previous child counterpart and the memory cell state $C_{t-1}$. The output of the LSTM has two outputs, the hidden output $h_t$ of the current word and its memory cell state $C_t$, respectively.

In the forgetting gate, the process is shown in Equation (13), where $f_t$ denotes the forgetting gate, $W_{fx}$, $W_{fh}$ are the weight matrices of the forgetting gate, and $\beta_f$ is the bias term of the forgetting gate. A number between 0 and 1 is obtained by the $\sigma$ output as the degree of forgetting of the previous long-term continued cell information. Similarly, the input gate $i_t$ and output gate $o_t$ are calculated in a similar way to the forgetting gate. The input gate calculation procedure is shown in Equation (14), and the output gate formula is shown in Equation (15):

$$f_t = \sigma(W_{fx}x_t + W_{fh}h_{t-1} + \beta_f) \tag{14}$$

$$\begin{aligned} i_t &= \sigma(W_{ix}x_t + W_{ih}h_{t-1} + \beta_i) \\ o_t &= \sigma(W_{ox}x_t + W_{oh}h_{t-1} + \beta_o) \end{aligned} \tag{15}$$

In the input gate, $W_{ix}$, $W_{ih}$ are the weight matrices of the input gate, $\beta_i$ is the bias term of the input gate, $W_{ox}$, $W_{ox}$ are the weight matrices of the output gate, and $\beta_o$ is the bias term of the output gate. The input gate determines the proportion of the state information generated by the current input $x_t$ that is updated into the current memory cell state $C_t$. The output gate determines the proportion of the current word's memory cell state after updating through the LSTM cell as the current hidden output. The state information generated by the current word input vector $x_t$ is calculated as shown in Equation (16):

$$g_t = \tanh(W_{C'x}x_t + W_{C'h}h_{t-1} + \beta_{C'}) \tag{16}$$

where $W_{C'x}$, $W_{C'h}$ are the corresponding weight matrices and $\beta_{C'}$ is the corresponding bias term. For the current long-term memory cell state $C_t$, it consists of two parts, one is the memory cell state information $c_{t-1}$ generated by the previous word, and the other is the state information $g_t$ generated by the current input word vector $x_t$, while the current memory cell state $C_t$ is calculated by Equation (17):

$$C_t = f_t \bullet C_{t-1} + i_t \bullet g_t \tag{17}$$

Finally, the obtained memory cell state $C_t$ is controlled using the output gate $o_t$ to calculate the output $h_t$ of the current hidden layer. The calculation process is shown in Equation (18):

$$h_t = o_t \bullet \tanh(C_t) \tag{18}$$

The $h_t$ calculated by the above process, through the action of the output gate, retains the information that needs to be remembered for a long time and removes the unimportant information, which can well express the dependencies between the input sequence words.

### 3.5.2. BiLSTM Model

The unidirectional LSTM model continuously updates the current state using the previous information but can only access the historical information of the upper context in the sequence and cannot be combined with the information of the lower context. However, in normal reading behavior, most of the time it is often necessary to combine the context to make sense of the whole sentence. Therefore, to address this problem, Dyer et al. [18] proposed a bidirectional long- and short-term memory model that can fully access information in both directions before and after the whole sequence. The structure diagram of the BiLSTM model is shown in Figure 9.

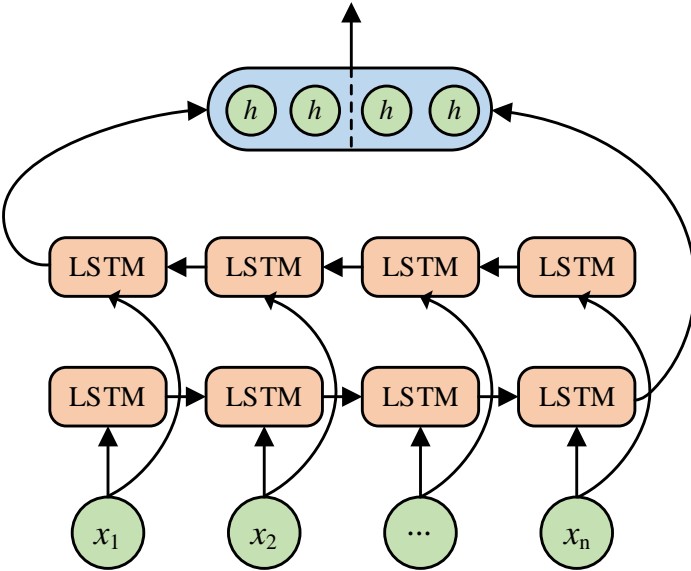

**Figure 9.** BiLSTM model structure.

The BiLSTM model does not change from the LSTM model in terms of structure. The BiLSTM model takes the sequence as well as the inverse sequence as input information and feeds them into the two one-way LSTM models, and combines the hidden vectors obtained in both directions as the output of the BiLSTM, which contains both the information of the preceding text in the utterance and considers the information of the following, resulting in a more accurate recognition of the named entities.

### 3.6. CRF Model

In the decoding task of named entity recognition, the commonly used decoding functions are SoftMax function and CRF. SoftMax function calculates the predicted labels of each word independently without considering the connection between two adjacent labels, so the decoded results of SoftMax function will show disorder and collocation error. CRF, on the other hand, can consider the relationship between the preceding and following annotations of tags from the sentence level, and can better solve the problems such as annotation bias that appear in SoftMax, so most named entity recognition tasks now use the CRF model as the decoder.

In the decoding stage, the label sequence computed by the feature extraction model is used as the input to the CRF model, which computes the conditional probability of the globally optimal output node. In CRF, let the transfer score matrix be $W$, $W_{y_i}$ denotes the score of transfer from $y_i$ label to $y_{i+1}$ label, and the score of the $i$-th element in the sequence at $y_i$ label is noted as $P_{i,y_i}$, then for a given input sequence $x = [x_1, x_2,..., x_n]$ of length $n$, the label sequence $y = [y_1, y_2,..., y_n]$, the specific calculation procedure of the score formula is shown in Equation (19):

$$\text{score}(x,y) = \sum_{i=1}^{n} P_{i,y_i} + \sum_{i=1}^{n+1} W_{y_{i-1},y_i} \tag{19}$$

Then, the probability distribution $P(y|x)$ of the input sequence $x$ and the label sequence $y$ is calculated by Equation (20):

$$P(y|x) = \frac{\exp(\text{score}(x,y))}{\sum\limits_{y' \in Y_X} \exp(\text{score}(x,y'))} \tag{20}$$

Finally, the maximum probability label is obtained based on normalization for the final annotation of the sentence.

## 4. Experiment and Analysis

### 4.1. Experimental Environment

The experiments were conducted on Ubuntu operating system with the following hardware information: Intel(R) Core (TM) i7-11700 @ 2.50 GHz, 16 GB of RAM, and 8 G of VRAM. This experiment uses the Python 3.6 programming language to build neural network models for training and testing using the deep learning architecture pytorch.

### 4.2. Experimental Data

The model proposed in this paper is also applicable in the English corpus, but this study mainly discusses the operation of the model under the Chinese corpus, so all Chinese corpus datasets are used.

The MSRA Chinese dataset and Weibo Chinese dataset, which are publicly available from Microsoft Asia Research Institute, are used for the experimental data. The MSRA dataset (https://github.com/InsaneLife/ChineseNLPCorpus/tree/master/NER/MSRA, accessed on 2 September 2022) contains the following entity types: organization name, place name, and person name [34]. The Weibo Chinese dataset (https://github.com/hltcoe/golden-horse, accessed on 2 September 2022) is generated by filtering and filtering the historical data of Sina Weibo between November 2013 and December 2014, and contains 1890 microblog mes-

sages [35]. The dataset entity categories are divided into four categories: person, organization, address, and geopolitical entity. The dataset settings are shown in Table 2.

**Table 2.** Data set settings.

| Data Set | Task | Training Set | Dev | Test Set |
|---|---|---|---|---|
| MSRA NER | Chinese NER | 73,001 | - | 7883 |
| Weibo NER | Chinese NER | 1350 | 270 | 270 |

*4.3. Labeling Strategy and Evaluation Index*

4.3.1. Labeling Strategy

Named entity recognition is mostly a sequence annotation task, the main work is to annotate the input text sequence with predefined tags. In the Chinese named entity recognition task, each word in the input Chinese text is assigned a tag. In the publicly available Chinese named entity corpus, the commonly used tagging patterns are BIO and BIOES.

In this paper, BIO is used as the labeling strategy, where B indicates that the current character is the first character of the entity name, I indicates that this character belongs to the non-first character of the entity name, and O indicates that this character is a non-named entity. By this method, seven kinds of labels to be predicted can be obtained by combining entity boundaries with entity types: "O", "B-PER", "B-LOC", "B-LOC "B-ORG", "I-PER", "I-LOC", and "I-ROG".

Based on the correctness of the output of the label format, consider the following three aspects:

1. The boundaries of the named entities are accurately delineated.
2. The labeling of named entities is classified correctly.
3. Each part of the named entity is labeled in an orderly manner.

If the labeling result does not meet the above criteria, the identification structure is considered incorrect.

4.3.2. Evaluation Indicators

Nested named entity recognition task as a special classification task, this paper uses precision rate $P$, recall rate $R$ and $F1$ value as experimental evaluation metrics to measure the recognition effect of the model.

$$P = \frac{T_P}{T_P + F_P} \times 100\% \tag{21}$$

where $T_P$ is the number of entities predicted correctly and $F_P$ is the number of entities predicted incorrectly. $T_P + F_P$ is the number of all entities identified.

Recall $R$ is the number of labels with correct annotation results that are labeled correctly, i.e., the proportion of the number of correct annotation results among the number of all named entities, and is calculated as shown in (22):

$$R = \frac{T_P}{T_P + F_N} \times 100\% \tag{22}$$

where, $F_N$ is the number of predicted non-entities. $F_P + F_N$ is the number of all labeled entities.

$F1$ is the weighted geometric mean of precision and recall, and the $F1$ value is the main indicator to judge the performance of the model, which is calculated as shown in (23):

$$F1 = \frac{2PR}{P + R} \times 100\% \tag{23}$$

where $R$ denotes the recall rate as described previously and $P$ denotes the precision rate as described previously.

### 4.4. Model Parameter Setting

This experiment uses the already trained BERT-base pre-trained model for vector representation of the input text, which has 12 layers, 12 heads of multi-headed attention, 768-dimensional hidden layer output, and 110 M parameter size. The parameters of the model are set as shown in Table 3.

**Table 3.** Model parameter setting.

| Parameters | Value |
|---|---|
| Max seq length | 128 |
| Learning rate | 0.0001 |
| Star dropout | 0.1 |
| LSTM dropout | 0.5 |
| CNN dropout | 0.5 |
| CNN filter height | 2, 3, 4, 5 |
| Star-Transformer layer | 1, 2, 3, 4 |
| Optimization functions | Adam |
| Loss function | Cross Entropy |
| Epoch | 20 |
| Batch size | 128 |

### 4.5. Experimental Results and Analysis

The precision, recall and F1 values of the model for each entity recognition when the maximum F1 is obtained are presented separately in Table 4.

**Table 4.** Identification results of different types of named entities.

| Type | MSRA | | | Weibo | | |
|---|---|---|---|---|---|---|
| | P (%) | R (%) | F1 (%) | P (%) | R (%) | F1 (%) |
| LOCAL | 94.74 | 94.16 | 94.45 | 60.85 | 63.41 | 62.10 |
| ORG | 94.63 | 93.81 | 94.22 | 61.86 | 62.57 | 62.21 |
| PER | 96.52 | 95.87 | 96.19 | 63.24 | 66.38 | 64.77 |
| Other | - | - | - | 62.73 | 63.87 | 63.29 |

The overall low experimental results of the Weibo dataset are attributed to the structure of the entities, because the data of the Weibo dataset are mainly from the Sina Weibo social platform, which contains entity types that may be structurally nested or informally written, so they cause greater interference to the prediction, resulting in poor prediction.

In order to verify the validity of the models, the following models are selected for comparison in this paper:

1. The BERT-BiLSTM-CRF model, which contains the current mainstream neural network models, is used as a baseline for comparison. The model uses a pre-trained BERT model to generate word vectors, which are input to BiLSTM-CRF for training.
2. The Radical-BiLSTM-CRF model, proposed by Dong et al. [36]. This model employs the BiLSTM-CRF network, and the word embeddings and strokes are input to the model for training.
3. The Lattice-LSTM-CRF model, proposed by Zhang et al. [37]. This model incorporates character and word granularity features at the embedding layer using an attention mechanism to encode the input character sequence and all potential words matched with the lexicon, where the word selection principle is that the character is located at the end of the word.
4. The CAN model, proposed by Zhu et al. [38]. This model uses a CNN with local attention and a bi-directional gated recursive unit (Bi-GRU) with global attention as an encoder and uses CRF for label prediction.

5. The TENER model, proposed by Yan et al. [39], which uses the Transformer encoder, introduces direction-awareness, distance-awareness, and un-scaled attention

6. BERT-CNN-BiLSTM-CRF, which uses a pre-trained BERT model to generate word vectors that are fed into a CNN-BiLSTM model for feature fusion and a CRF for label prediction.

The results of the comparison of precision, recall and F1 values between different models are shown in Tables 5 and 6.

**Table 5.** Comparison of the results of each model (MSRA).

|   | Models | MSRA | | |
|---|--------|------|---|---|
|   |        | **P (%)** | **R (%)** | **F1 (%)** |
| 1 | BERT-BiLSTM-CRF | 92.67 ± 0.26 | 91.73 ± 0.22 | 92.20 ± 0.23 |
| 2 | Radical-BiLSTM-CRF | 90.13 ± 0.31 | 88.17 ± 0.29 | 89.14 ± 0.29 |
| 3 | Lattice-LSTM-CRF | 91.39 ± 0.34 | 90.94 ± 0.45 | 91.16 ± 0.35 |
| 4 | CAN | 93.53 ± 0.25 | 92.24 ± 0.38 | 92.88 ± 0.29 |
| 5 | TENER | 92.28 ± 0.28 | 91.36 ± 0.41 | 91.82 ± 0.32 |
| 6 | BERT-CNN-BiLSTM-CRF | 94.13 ± 0.24 | 93.79 ± 0.40 | 93.96 ± 0.22 |
| 7 | Our | 95.13 ± 0.21 | 94.44 ± 0.35 | 94.78 ± 0.28 |

**Table 6.** Comparison of the results of each model (Weibo).

|   | Models | Weibo | | |
|---|--------|-------|---|---|
|   |        | **P (%)** | **R (%)** | **F1 (%)** |
| 1 | BERT-BiLSTM-CRF | 56.76 ± 0.31 | 63.87 ± 0.27 | 60.11 ± 0.28 |
| 2 | Radical-BiLSTM-CRF | 54.94 ± 0.41 | 62.48 ± 0.31 | 58.47 ± 0.33 |
| 3 | Lattice-LSTM-CRF | 53.17 ± 0.36 | 62.25 ± 0.38 | 57.35 ± 0.35 |
| 4 | CAN | 55.68 ± 0.30 | 63.21 ± 0.32 | 59.21 ± 0.33 |
| 5 | TENER | 55.32 ± 0.22 | 62.73 ± 0.45 | 58.79 ± 0.34 |
| 6 | BERT-CNN-BiLSTM-CRF | 56.24 ± 0.43 | 62.28 ± 0.33 | 59.11 ± 0.36 |
| 7 | Our | 61.36 ± 0.32 | 62.77 ± 0.29 | 62.06 ± 0.29 |

The comparison results show that the proposed model achieves 94.78% and 62.06% F1 values on the MSRA and Weibo datasets, respectively. Additionally, on the MSRA dataset, the model proposed in this paper is higher than the other comparison models in terms of precision, recall and F1 value.

Comparing 1 with 2, 3 and 4, it can be found that model 1 has the highest F1 value, which indicates that BERT is richer in extracting features than training stroke features and character fusion features alone, and the BERT word vector better combines the context and can better represent the semantic information of words.

Comparing 4 and 5, it can be found that model 4 performs better than the Transformer model which encodes character features by performing fusion of local and global features, indicating that fusion of local and global features can better improve the recognition effect.

Comparing 6 and 7, we can find that the same BERT model is used for preprocessing and both local and global features are fused, and the F1 value of model 7 is higher than that of model 6 in both datasets, which indicates that the Star-Transformer-CNN-BiLSTM fusion used in this paper is richer in features.

### 4.6. Ablation Experiments

In order to verify the validity of each part of our proposed model, we designed the ablation experiment.

The data set for the ablation experiment is MSRA, and the data set settings are the same as those for the comparison experiment, and the ablation experiment settings are as follows:

1. BiLSTM: use the BiLSTM model instead of the CNN-BiLSTM model.
2. CNN: use CNN model instead of CNN-BiLSTM model.
3. CNN-BiLSTM: remove the CNN-BiLSTM model.
4. Transformer: use the Transformer model to replace the Star-Transformer model.

The results of the ablation experiments are shown in Table 6.

It can be found from Table 7 that the F1 values of BiLSTM model, CNN model, CNN-BiLSTM model and Transformer model have decreased. Because the CNN can only acquire local features, the F1 value of CNN model decreases by 5.96%. The BiLSTM model can acquire global features, but lacks the fusion with local features of CNN, which decreases the F1 value by 1.26%. The CNN-BiLSTM model decreases the F1 value by 7.15% because it only uses the Star-Transformer model for local feature extraction. Although the Transformer model obtained richer feature information by fusing local features with global features, the F1 value decreased by 0.31%, which indicates that the feature extraction ability of Transformer is slightly lower than that of Star-Transformer.

**Table 7.** Results of ablation experiments.

| Models | P (%) | R (%) | F1 (%) | ΔF1 (%) |
|---|---|---|---|---|
| Our | 95.13 | 94.44 | 94.78 | - |
| BiLSTM | 93.16 | 93.89 | 93.52 | −1.26 |
| CNN | 90.21 | 87.47 | 88.82 | −5.96 |
| CNN-BiLSTM | 89.18 | 86.13 | 87.63 | −7.15 |
| Transformer | 94.64 | 94.31 | 94.97 | −0.31 |

After the ablation experiments, it can be shown that the model proposed in this paper has better feature extraction ability, and it shows that improving the information representation of characters by fusing related entity information can effectively improve the recognition effect of named entities.

### 4.7. Computational Efficiency Experiments

In order to verify the effectiveness of the Star-Transformer model in terms of computational efficiency, the following two models are selected for comparison experiments.

BERT-Transformer-CNN-CRF: the word vector is generated using the already pre-trained BERT model as the encoding layer, and then the word vector is input to the joint model of Transformer and CNN for feature extraction and fusion, and finally decoded using the CRF layer.

BERT-Star-Transformer-CNN-CRF: the design idea is the same as the BERT-Transformer-CNN-CRF model, except that the Transformer model is replaced by the Star-Transformer model to verify the experimental effect between the two models.

The dataset for the efficiency experiments is the same as the model comparison experiments, and the same MSRA Chinese dataset is used for the experiments, and the dataset settings are the same as the model comparison experiments.

#### 4.7.1. Model Parameter Setting

The parameter settings of the two models are shown in Table 8.

**Table 8.** Model parameter setting.

| BERT-Transformer-CNN-CRF | | BERT-Star-Transformer-CNN-CRF | |
|---|---|---|---|
| **Parameters** | **Value** | **Parameters** | **Value** |
| Max seq length | 128 | Max seq length | 128 |
| Learning rate | 0.0001 | Learning rate | 0.0001 |
| Transformer dropout | 0.4 | Star dropout | 0.1 |
| CNN dropout | 0.5 | CNN dropout | 0.5 |
| epoch | 20 | epoch | 20 |

#### 4.7.2. Efficiency Experimental Results and Analysis

The results of the comparison experiments are shown in Figure 10. The comparison shows that the precision, recall and F1 values of the BERT-Star-Transformer-CNN-CRF

model are higher than those of the BERT-Transformer-CNN-CRF model, where the difference in precision is 0.91%, the difference in recall is 0.69% and the difference in F1 is 0.79%. This indicates that the Star-Transformer-CNN model has a better feature extraction capability than the Transformer-CNN model.

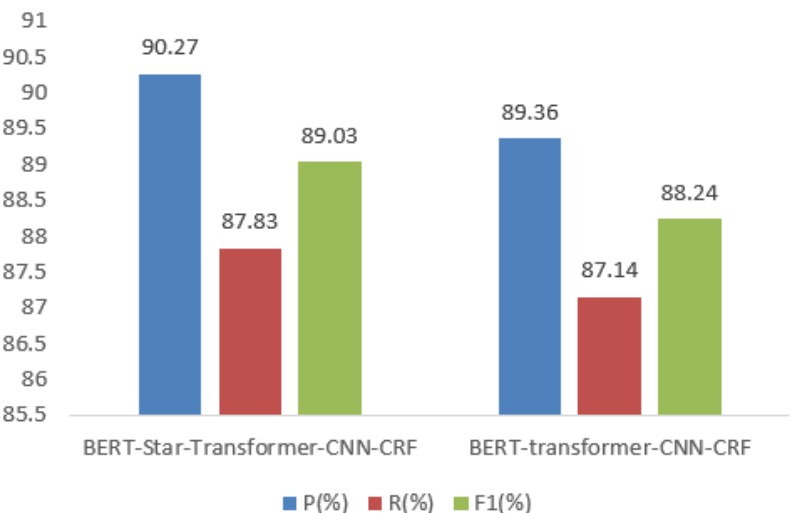

**Figure 10.** Comparison of experimental results.

The comparison results of F1 values between the BERT-Star-Transformer-CNN-CRF model and the BERT-Transformer-CNN-CRF model at each iteration cycle are shown in Figure 11. The comparison shows that the BERT-Star-Transformer-CNN-CRF model converges faster and is able to reach a higher F1 value in the early training period, and continues to improve and finally maintains at a higher level. While the BERT-Transformer-CNN-CRF model has a faster convergence speed after the first five iterations, it only obtains a higher value after many iterations, but never exceeds the BERT-Star-Transformer-CNN-CRF model.

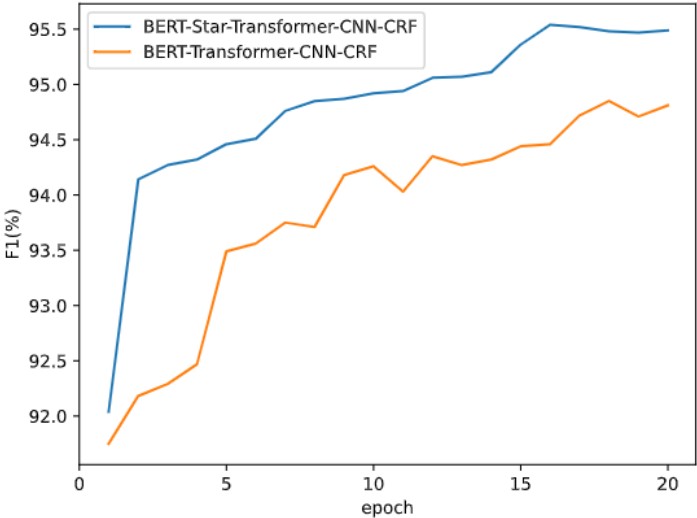

**Figure 11.** F1 value change comparison.

The Transformer model requires a longer training time than the Star-Transformer model in terms of computational time because of its internal fully connected structure, so the computational overhead is too large.

The final training time for the BERT-Transformer-CNN-CRF model is 61,857 s, while the final training time for the BERT-Star-Transformer-CNN-CRF model has a final training

time of 25,136 s. In terms of computational efficiency, the BERT-Star-Transformer-CNN-CRF model an improvement 40.6% greater than the BERT-Transformer-CNN-CRF model. The training times of the two models are shown in Table 9.

**Table 9.** Training time for each epoch.

| Epoch | BERT-Transformer-CNN-CRF Training Time (s) | BERT-Star-Transformer-CNN-CRF Training Time (s) |
|---|---|---|
| 1 | 3968 | 1286 |
| 2 | 6923 | 2478 |
| 3 | 9916 | 3675 |
| 4 | 12,876 | 4868 |
| 5 | 15,786 | 6064 |
| 6 | 18,762 | 7254 |
| 7 | 21,731 | 8456 |
| 8 | 24,728 | 9655 |
| 9 | 27,739 | 10,848 |
| 10 | 30,736 | 12,045 |
| 11 | 33,724 | 13,249 |
| 12 | 36,719 | 14,443 |
| 13 | 39,706 | 15,637 |
| 14 | 42,727 | 16,841 |
| 15 | 45,726 | 18,037 |
| 16 | 48,728 | 19,238 |
| 17 | 51,739 | 20,442 |
| 18 | 54,725 | 21,641 |
| 19 | 57,683 | 22,836 |
| 20 | 60,692 | 24,029 |

## 5. Conclusions

To solve the problem of multiple meanings of the word in Chinese named entity recognition tasks, this paper proposes a deep learning model based on the BERT pre-training model, which uses a lightweight Star-Transformer model in feature extraction instead of the traditional Transformer model, because the Star-Transformer changes the internal structure compared with the traditional Transformer model. The Star-Transformer changes the internal fully connected structure, which improves the computational efficiency of the model, and uses the mainstream CNN-BiLSTM joint model to extract the feature sequences and fuse them with the feature sequences obtained from the Star-Transformer model, which fully takes into account the semantic relationship of the context in the sentence and further alleviates the problem of multiple meanings of the word in Chinese utterances, and finally the CRF is used in decoding to obtain the best label discrimination results. After the comparison of experimental results, it is proven that the present model has better performance in the recognition of Chinese named entities. Although the present model uses a lightweight Star-Transformer to improve the computational efficiency, the BiLSTM cannot use the parallelism of GPU, it leads to the shortage of computational efficiency, so our future work focuses on finding a lightweight model to extract the global feature information.

After years of efforts by scholars, the recognition rate of named entities in the general domain has reached a high level, but in some specific domains, such as the medical domain, the architecture domain, the news domain and other specific domains, the task of named entity recognition in these special domains still has a large number of challenges due to the complex naming rules of named entities in these special domains. This is also the focus of our future work.

**Author Contributions:** Conceptualization, Y.G.; Methodology, Y.G. and R.Y.; Software, R.Y.; Validation, R.Y. and C.Z.; Resources, C.Z.; Writing—original draft preparation, R.Y.; Writing—review and editing, R.Y. and C.Z. All authors have read and agreed to the published version of the manuscript.

**Funding:** This research was funded by the Nation Nature Science Foundation of China (NSFC), (NO. 61572445).

**Institutional Review Board Statement:** Not applicable.

**Informed Consent Statement:** Not applicable.

**Data Availability Statement:** The data presented in this study could be provided by request.

**Conflicts of Interest:** The authors declare no conflict of interest.

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
