# Peer review of "Chinese Named Entity Recognition Based on BERT and Lightweight Feature Extraction Model"

_information, doi:10.3390/info13110515_

Round 1

Reviewer 1 Report (Previous Reviewer 2)

# Second review

Below, how I rate the adjustments made that I asked for. I find them insufficient unfortunately. 

## Suggestions for improvements

1. Add researcgh questions

    2.  not added.

2. Indicate which aspect of this research is new. What is the research gap this paper fills.

    3. I could not find it in the manusscript. Th eterms "gap" and "novel" do not appear.

3. Provide reference to the dataset in 4.2

    4. insufficient: a URL to the exact two sets is needed. 

4. Make it **exactly clear how predictions and ground truth are matched** in 4.3.2. As it is now it is much too vague.

    5. insufficient, is exact match or partial match used? 

5. provide confidence intervals in tables 4 and 5

    6. insufficient, also which the few CI provided, and that high, it is not right to give the scores up to 1/10.000 precision.

6. Do not just compare to your own work, but also to scores obtained by others on exactly this dataset. What are the sota baselines that you want to b eat?

    7. This is a great addition, but without c-i's  somewhat meaningless, as sota  system might still perform the same as th eproposal in the paper.

Author Response

Reviewer 2 Report (New Reviewer)

This paper aims to improve the semantic word representation in the Chinese named entity recognition. The proposed method utilizes well-known NLP modules such as BERT, Transformer, CNN, BiLSTM, and CRF. Transformer has been widely used for name entity recognition. But instead of using Transformer the proposed method adopts Star-Transformer, which was proposed to reduce computational and memory overheads, to devise a light-weight NER model. The readability of the paper should be further; The uncomplete manuscript seems to be submitted because it still has red highlights. Furthermore, I found that it includes repeated sentences.

Comments:

- The manuscript includes too many irrelevant backgrounds such as Transformer encoder unit.

- In Table 4, the proposed model shows the highest precision score, but the lowest recall score in Weibo dataset. The more discussion is helpful to understand the role of Star-Transformer module.

- Provide more extensive experiments to show if the semantic word representation actually is improved by the proposed technique.

- Experiment 4.7 verifies the validity of Star-Transformer using the subnetwork of proposed architecture (without BiLSTM module). But here, two concerns arise. First, without BiLSTM model shows better performance in F1 score (Figure 10, 95.54; Table 4, 95.26). Second, the training time comparison does not guarantee the proposed model's improvement.

- Minor issues

  - Please indicate the source of pretrained BERT.

- Some equations are not well-defined. e.g., Eq 16 (g_t -> C_t’), 17 (C_t -> C_{t-1})

- In line 579, Table 6 -> Table 5

  - Some lines are incomplete. e.g., line 96 (incomplete sentence), 641 (duplicated sentence)

  - In Table 5, some records don't make sense. e.g., -CNN-BiLSTM model

Round 2

Reviewer 2 Report (New Reviewer)

All comments are well addressed in the new submission.

This manuscript is a resubmission of an earlier submission. The following is a list of the peer review reports and author responses from that submission.

Round 1

Reviewer 1 Report

This paper proposes a BERT-Star-Transformer-CNN-BiLSTM-CRF model for Chinese NER. The authors combine several popular model architectures together and show good evaluation results on one Chinese NER dataset. Following are my main concerns:

- The experiments and comparisons are not enough. Only one Chinese NER dataset is used in evaluation. Moreover, the proposed model is only compared with some of its variants, there is no rigorous comparison with other publications. For example, this paper (https://aclanthology.org/2022.findings-naacl.143.pdf) listed several methods use the same dataset. It also evaluates on OntoNotes. 

- It is unclear why the proposed method is for Chinese NER. It seems to me the approach can work for English NER as well. It would be good to provide some clarification on this. 

- Since the proposed approach is quite complex, it would be good to conduct ablation study to show the usefulness of each component. For example, what's the impact of removing the RNN component while keeping all the other components?

- The writing and presentation can be greatly improved. There are many missing citations and types. Moreover, many sentences are not finished. 

Following are some detailed comments:

  • Line 45: sentence cutoff after “low”
  • Line 44: it’s not totally true that lexicon method cannot handle new names. There are features try to generalize from these lexicons
  • Type in Figure 1? Ster -> Star
  • Line 194: sentence cutoff after “to get”
  • Table 1 seems to be to wide, which might not follow the requirements of paper layout
  • Line 238: please cite the study
  • Line 274: should it be K instead of V?
  • Section 3.3: there is no citation of Star-Transformer

Reviewer 2 Report

# Review of Chinese Named Entity Recognition Based on BERT and Light- 2 weight Feature Extraction Model

Well written paper that shows impressive results on a chinese NER dataset using a system which combines all SOTA techniques.

## Suggestions for improvements

1. Add researcgh questions

2. Indicate which aspect of this research is new. What is the research gap this paper fills.

3. Provide reference to the dataset in 4.2

4. Make it **exactly clear how predictions and ground truth are matched** in 4.3.2. As it is now it is much too vague.

5. provide confidence intervals in tables 4 and 5

6. Do not just compare to your own work, but also to scores obtained by others on exactly this dataset. What are the sota baselines that you want to b eat?
